# Emerging Landscape of Immunotherapy for Primary Central Nervous System Lymphoma

**DOI:** 10.3390/cancers13205061

**Published:** 2021-10-10

**Authors:** Marion Alcantara, Jaime Fuentealba, Carole Soussain

**Affiliations:** 1Center for Cancer Immunotherapy, Institut Curie, PSL Research University, INSERM U932, 75005 Paris, France; marion.alcantara@curie.fr (M.A.); jaime.fuentealba@curie.fr (J.F.); 2Clinical Hematology Unit, Institut Curie, 92210 Saint-Cloud, France

**Keywords:** CNS lymphoma, immunotherapies, brain tumor microenvironment

## Abstract

**Simple Summary:**

Primary central nervous system lymphoma (PCNSL) is characterized by its location in the central nervous system comprising the brain, the eye, the cerebrospinal fluid and the spinal cord and a poor prognosis with the current chemotherapies. Immunotherapies represent a new paradigm in the care of patients with B-cell lymphoma, but, till recently, immunotherapies studies excluded patients with PCNSL because of the lack of knowledge on the immune network in the brain. Recent studies shed a new light on the origin and characteristics of the CNS immune cells. We review the current experimental preclinical and clinical developments of immunotherapies in CNS lymphoma as well as the effects of targeted therapies on the brain microenvironment. We provide perspectives for improving the efficacy of immunotherapies in the specific setting of PCNSL for a better prognosis of this disease.

**Abstract:**

Primary central nervous system lymphoma (PCNSL) is, mainly, a diffuse large B-cell lymphoma (DLBCL) with a non-germinal center B-cell (non-GCB) origin. It is associated with a poor prognosis and an unmet medical need. Immunotherapy has emerged as one of the most promising areas of research and is now part of the standard treatment for many solid and hematologic tumors. This new class of therapy generated great enthusiasm for the treatment of relapsed/refractory PCNSL. Here, we discuss the challenges of immunotherapy for PCNSL represented by the lymphoma cell itself and the specific immune brain microenvironment. We review the current clinical development from the anti-CD20 monoclonal antibody to CAR-T cells, as well as immune checkpoint inhibitors and targeted therapies with off-tumor effects on the brain microenvironment. Perspectives for improving the efficacy of immunotherapies and optimizing their therapeutic role in PCNSL are suggested.

## 1. Introduction

Primary central nervous system lymphomas (PCNSL) involve the brain, the eye, the cerebrospinal fluid (CSF) and, less frequently, the spinal cord, without any systemic dissemination. In immunocompetent patients, the histology is almost always a diffuse large B-cell lymphoma (DLBCL), preferentially of a non-germinal center (non-GC) phenotype [1,2]. PCNSL is a rare disease, representing 3% of all non-Hodgkin lymphomas (NHL), with no specific sex ratio and affecting 1900 and 300 new cases per year in the USA and France, respectively [1,3]. The prognosis of PCNSL is poorer than that of nodal non-GC DLBCL [4]. Improved outcomes have been observed in recent decades, especially in young patients who respond to high-dose methotrexate-based induction chemotherapies and who receive a consolidation treatment with either intensive chemotherapy and autologous stem cell transplantation (IC-ASCT) or whole brain radiation therapy (WBRT) [5,6]. IC-ASCT acts on minimal residual disease through the dose-intensity effect of the chemotherapy. IC-ASCT showed a good control of the disease in the first-line setting [5,6]. At relapse, consolidation with IC-ASCT is an effective treatment, allowing a survival gain in patients under the age of 65 years who are eligible for such an intensive treatment [7,8]. However, IC-ASCT exposes to the risk of IC-related toxicities and treatment-related deaths in 4 to 10% of the patients [5,6,7].

Despite these therapeutic improvements, 16 to 26% of patients are primary refractory to high-dose methotrexate [1,9] or, subsequently, relapse [10]. These patients represent an unmet medical need [1,8].

The poor prognosis of PCNSL with the current conventional treatments can be, at least in part, explained by the aggressiveness of non-GC malignant B-cells, with frequent mutations of *MYD88* and *CD79b* [4], and by the anatomical and functional characteristics of the blood–brain barrier (BBB) which limits the bioavailability of many drugs in the brain parenchyma. The homing of PCNSL in the CNS at diagnosis is also observed at relapse, with less than 5% of relapses occurring outside the CNS in a series of 1000 patients in the modern era [1]. This points out the role of the tumor microenvironment (TME) as a key component of lymphomagenesis and homing in the CNS [11,12,13] and, therefore, as a therapeutic target to be explored.

Immunotherapies represent a new paradigm in the care of patients with systemic B-cell malignancies. The benefit of immunotherapies, such as allogeneic hematopoietic stem cell transplantation, monoclonal anti-CD20 antibodies, immune checkpoint inhibitors and CAR-T cells, have been less explored in PCNSL, because of the rarity of the disease and the concerns raised by the expected lack of immune effector cells in the brain. Indeed, the brain, while not being an immune-privileged sanctuary, still provides an immunosuppressive and nurturing tumor environment, which could provide resistance to immunotherapies. The clinical activity of ibrutinib, lenalidomide and pomalidomide has been demonstrated in relapsed PCNSL. How these drugs, known to modulate the microenvironment of B-cell malignancies, impact the immune brain microenvironment remains to be deciphered. 

This review aims to present an overview of the knowledge regarding the immune brain microenvironment, the most important preclinical and clinical results, along with suspected underlying mechanisms of resistance, and provide perspectives for improving the efficacy of immunotherapies and optimizing their role in the therapeutic armamentarium of PCNSL.

## 2. Brain Microenvironment

In recent years, immunotherapy has become increasingly common in the management of solid tumors and some B-cell lymphomas due to clear clinical benefits. Immunotherapy efficacy depends on several aspects that are tumor specific. Both autonomous mechanisms (e.g., the low tumor mutational burden, downregulation of MHC genes and expression of PD-L1/PD-L2), as well as extrinsic properties (e.g., immunosuppressive TME), are at the origin of inconsistent clinical outcome. Although the TME of certain brain tumors is starting to be unraveled, little is known about the cellular and molecular immune players implicated in PCNSL progression. PCNSL can develop in the brain parenchyma, but also in CNS interfaces: the perivascular and meningeal spaces. It is expected that the precise location of PCNSL will drive the composition of the TME (Figure 1).

The CNS has been historically considered to be immune-privileged. This concept has been coined following Medawar’s observation that skin allografts have a prolonged survival when placed in the brain and eye of rabbits [14]. The inability to mount an efficient immune response against the grafts was explained by the presence of physical barriers at the borders between the CNS and the periphery: the blood–brain barrier (BBB) and the blood–cerebrospinal fluid barrier (BCSFB), as well as the apparent lack of lymphatic drainage. Nowadays, the concept of immune privilege has been revisited, due to the better understanding of CNS immunity. Recently, the presence of lymphatic vessels was found in the meninges of mammals. The network of lymphatic vessels runs parallel to dural venous sinuses and allows for the drainage of cells and CSF into deep cervical lymph nodes [15,16,17]. Although the brain does not seem to be directly drained, interstitial fluid (ISF) solutes are constantly being cleared and carried into the CSF through the “glymphatic system”. Instead of lymphatic vessels, the glymphatic system uses periarterial spaces to move CSF into the brain parenchyma and perivenous spaces to drive ISF out [18]. In the steady state, the immune system of the CNS is composed mainly of innate immune cells. These cells are mostly macrophages found in the parenchyma, namely, microglia, but also in the borders of the CNS: meningeal, perivascular and choroid plexus macrophages [19]. Fate-mapping experiments combined with transcriptomic studies have shown that CNS macrophages originate from yolk sac erythro-myeloid progenitors and are believed to be sustained by self-renewal during adulthood [19]. Very recently, the existence of direct vascular connections between the meninges and the skull bone marrow was described in mice. It was shown that the skull and the vertebral bone marrow represent a private reservoir of myeloid cells and B-cells that replenishes the meninges, perivascular spaces and the CNS parenchyma during homeostasis and CNS injury [20,21,22], in complement to blood-derived inflammatory cells (Figure 1). If such reservoirs also exist in humans, it will reshape our interpretation of CNS immunity in health and disease. In light of these findings, it is tempting to speculate that a dural bone marrow reservoir of B-cells might be involved in PCNSL pathophysiology and its exclusive homing to the CNS.

Although a more detailed characterization of the TME in large cohorts of patients is missing, several studies have attempted to identify cellular and molecular mechanisms implicated in the progression of PCNSL. The presence of tumor-infiltrating lymphocytes (TILs) has been observed [13,23,24,25,26], and its accumulation within the perivascular space has been associated with better survival [23]. Some TILs express immune checkpoint receptors (i.e., PD-1 and TIM3) [25,26,27] and a high expression of PD-1 has been associated with inferior survival [28]. Besides lymphoma cells, tumor-associated macrophages (TAMs) have been described as alternative sources of PD-L1 [26,29]. TAMs are a mixed population of macrophages with different ontogenies (microglia, perivascular/meningeal macrophages and monocyte-derived macrophages) and a global increased ratio of M1/M2-like TAMs has been associated with a better outcome [27]. Furthermore, the level of infiltration of TAMs has been correlated with IL-10 in the CSF [30]. IL-10 has been mostly reported as an anti-inflammatory cytokine, which plays a central role in lymphoma development as a growth factor for B-lymphocytes and an inducer of the anti-apoptotic Bcl2 pathway [31]. IL-10 is produced by the lymphoma cells and could have both autocrine and paracrine effects. IL-10 was identified as an effective diagnostic biomarker for PCNSL [32,33]. Finally, besides cells derived from direct immune lineages, other cell populations including astrocytes, mural cells and endothelial cells are likely involved in shaping up the immune landscape of PCNSL TME [13,34].

In the era of single-cell omics, we should soon be able to better understand the complexities of the TME of PCNSL; consequently, boosting the development of more effective immune-based therapies.

## 3. Available Clinical Data

### 3.1. Allogeneic Hematopoietic Stem Cell Transplantation (alloHSCT)

From a historical perspective, alloHSCT represents the first success of immunotherapy [35]. Assuming that allogeneic T-cells may traffic to the CNS and mediate the graft-versus-lymphoma (GvL) immunoreaction, alloHSCT may be effective against PCNSL [36]. The GvL immunoreaction, specific to alloHSCT, is exclusively sought if a non-myeloablative conditioning regimen is chosen, or will complete the effect of the intensive chemotherapy on the minimal residual disease if a myeloablative regimen is given. The conditioning regimen’s choice is driven by the patients’ characteristics and previous treatments. Nevertheless, allogeneic T-cells can lead to graft-versus-host disease (GvHD), which can be severe and life-threatening. Mika et al. conducted a retrospective study on 6 PCNSL patients who received alloHSCT following a conditioning regimen with fludarabine, busulfan and cyclophosphamide [37]. All patients had previously received rituximab and high-dose methotrexate, in combination with high-dose cytarabine or ifosfamide, as first-line therapy. All patients had also received IC-ASCT, three of whom in first-line consolidation and the others in second-line. All patients presented with an unconfirmed complete response (uCR) before alloHSCT. Two patients died, one from severe GvHD and one from a lymphoma relapse. Four out of six patients were still alive in complete response (CR) almost 4 years after alloHSCT. Although preliminary, these data suggest that immune mechanisms might be active against PCNSL (Table 1). AlloHSCT may be an interesting option after failure of IC-ASCT. Ideally, prospective trials would help to unravel this therapeutic potential, together with safety considerations, and to better define its position regarding other cell therapies, such as chimeric antigen receptor (CAR) T-cells [38].

### 3.2. Monoclonal Antibodies

The CD20-directed monoclonal antibody, rituximab, has shown survival improvement in B-cell non-Hodgkin lymphomas (NHL), including systemic DLBCL [39], and is now part of their gold standard therapies [40]. Considering that most of PCNSL are CD20+ subtypes of DLBCL [41], it makes sense hypothesizing that rituximab would improve the outcome of PCNSL patients. However, the addition of rituximab in the treatment of PCNSL is not straightforward. Under physiologic conditions, the BBB prevents the trans-vascular crossing of most molecules larger than 180 Daltons [42]. Nonetheless, the alteration of the BBB observed around large tumor lesions in PCNSL may improve the penetration of monoclonal antibodies.

From a preclinical perspective, rituximab has shown an activity following intravenous injection in a nude rat model of CNS lymphoma [43]. From a clinical perspective, four confirmed radiographic responses (3 CR, 1 partial response (PR)) were reported in a pilot study including twelve patients with recurrent or refractory PCNSL who received rituximab as monotherapy [44]. Promising results have also been reported in retrospective studies, showing that rituximab improves CR rates, progression-free (PFS) and overall survival (OS), in combination with high-dose chemotherapy [45,46]. However, this benefit was challenged by the HOVON/ALLG international, multicentric, randomized phase 3 trial [47] (Table 1). One hundred newly diagnosed PCNSL patients received two cycles of MBVP (methotrexate, carmustine, teniposide and prednisone) and ninety-nine patients received the same induction regimen, in combination with intravenous rituximab on days 0, 7, 14 and 21 in cycle one and days 0 and 14 in cycle two. Patients in response (CR or PR) at the end of induction received a consolidation with high-dose cytarabine and, for patients aged 60 years or younger, low-dose WBRT. The authors found no difference, neither in the terms of event-free survival (EFS, primary endpoint of the study, 49% versus 52% at 1-year) nor PFS, OS or response to induction chemotherapy. An unplanned subgroup analysis showed a trend for enhanced EFS for patients aged 60 years or younger who received rituximab. Another international, multicentric trial aimed at evaluating the effects of rituximab in first line PCNSL [9]. This phase two, IELSG-32 trial, compared three different induction chemotherapy regimens in a randomized manner, i.e., high-dose methotrexate (HD-MTX) plus cytarabine (*n* = 75 patients), HD-MTX, cytarabine plus rituximab (*n* = 74 patients), and HD-MTX, cytarabine and rituximab plus thiotepa (*n* = 78 patients). A second randomization compared consolidation with WBRT or ASCT for patients in response or stable disease. The primary endpoint of the first randomization was the CR rate after four cycles of induction therapy, which was 23%, 30% and 49% in each of the three arms, respectively. A systematic review and meta-analysis including these two randomized trials, for a total of 343 patients, was conducted to provide guidance for clinical practice [48]. Albeit, neither trial demonstrated a benefit for rituximab regarding their primary endpoints, the pooled hazard ratio for PFS suggested a possible benefit of adding rituximab. This meta-analysis also showed that the addition of rituximab was not associated with an increase in clinically significant adverse events. The long-term analysis of the IELSG-32 study was recently presented at the ICML 2021 meeting and showed a better 7-year OS for patients who received HD-MTX, cytarabine plus rituximab (37%) compared to patients who received only HD-MTX plus cytarabine (26%) independently of the consolidation arm [49].

Altogether, the addition of rituximab to HD-MTX-based chemotherapy did not dramatically improve the prognosis of patients with PCNSL as observed in systemic DLBCL. Despite modest evidence, rituximab has been mostly integrated into PCNSL first and subsequent lines of therapy. Of note, to our knowledge, no data regarding the activity of bispecific antibodies in PCNSL are available to date.

### 3.3. Immune Checkpoint Inhibitors (ICI)

Checkpoint blockade using programmed cell death (ligand) 1 (PD-(L)1) antibodies made a huge breakthrough in the treatment of many tumor types previously limited by the lack of therapeutic options [50,51,52,53]. Particularly, anti-PD-1 antibodies showed dramatic anti-tumor responses in classical Hodgkin lymphoma (HL) [54,55], which is characterized by chromosome 9p24.1 alterations, including polysomy, copy gain and amplification, and resulting in enhanced PD-L1/PD-L2 expression [56]. High-level 9p24.1 copy gain and increased PD-L1 expression are associated with prolonged survival of HL patients upon anti-PD-1 treatment [57]. Interestingly, 9p24.1/PD-L1/PD-L2 copy number alterations and translocations of these loci were reported in more than 50% of EBV-negative PCNSL [4]. These structural bases for PCNSL immune evasion, together with the characterization of the tumor microenvironment [27], support the use of ICI in PCNSL. Preliminary encouraging results have been reported with the anti-PD-1 antibody, nivolumab single agent, in four patients with relapsed/refractory (R/R) PCNSL and one patient with CNS relapse of testicular lymphoma [58]. Overall, nivolumab was well tolerated. All patients had objective responses, including 4 CR and 1 PR, and three patients remained progression-free at 13 to 17 months. Nonetheless, two patients had received radiation therapy immediately prior to the initiation of nivolumab. A high objective response rate was also reported in another monocentric retrospective study, including eight PCNSL patients treated with nivolumab. Three and four patients achieved a CR and a PR, respectively [59]. Unfortunately, these promising results have not been confirmed in a prospective study including 47 PCNSL and 19 patients with CNS relapse of primary testicular lymphoma (NCT02857426), according to the available results posted on clinicaltrials.gov in 2020. These results have not been published yet. Pembrolizumab has also been studied in R/R PCNSL. The first results of the AcSé pembrolizumab multicentric phase II study were presented at the ASH 2020 annual meeting [60] (Table 1). Fifty R/R patients, including 41 PCNSL and 9 primary vitreoretinal lymphomas (PVRL), treated with single agent pembrolizumab were reported. Eight and five patients obtained a CR and a PR, respectively, leading to an overall response rate of 26% and a median PFS of 2.6 months. Responses may be durable as the reported median duration of response (DOR) was 10 months. Considering the good safety profile, further studies evaluating ICI either in combination therapies and/or earlier in the course of the disease are warranted to increase their activity in PCNSL.

### 3.4. CAR-T Cells

CAR-T are genetically engineered T-cells that express an antibody-like chimeric receptor [61]. Autologous CAR-T targeting CD19 are currently approved and commercialized in third-line systemic DLBCL [62,63,64]. Nevertheless, few data on CAR-T for CNS lymphoma are available to date [38] and most of them concern secondary, but not primary, CNS lymphoma. Patients with CNS infiltration were excluded from most of the pivotal trials because of concerns related to immune effector cell-associated neurotoxicity syndrome (ICANS) after CAR-T therapy [65]. Patients with secondary CNS involvement were eligible for the TRANSCEND prospective study, which aimed to assess the safety and activity of liso-cel in third-line DLBCL [64]. Among 256 evaluable patients, 6 had CNS disease and 3 of them achieved a CR. The largest cohort of secondary CNS lymphoma patients was published by Frigault et al., who reported a retrospective analysis on eight patients who received tisa-cel [66]. The treatment was well tolerated and responses were observed in four patients (2 CR, 2 PR) at day 28 after CAR-T infusion. Li et al. showed results with a longer follow-up for five patients, including one primary and four secondary CNS lymphomas, enrolled in a clinical trial testing CD19 plus CD22 CAR-T cells [67]. All achieved an objective response, but four patients relapsed within 3 to 8 months. The authors suggested that the immunosuppressive brain microenvironment may have contributed to the lymphoma relapse. This interesting hypothesis should be addressed by relevant preclinical and clinical studies. For example, preclinical experiments, including histopathology, flow cytometry and single-cell RNA sequencing, in immunocompetent animals, could assess the role of the brain microenvironment in the relapse following CAR-T therapy. From a clinical perspective, combining CAR-T cells with immunomodulatory agents seems relevant. Recently, the French national network for oculo-cerebral lymphomas (LOC) reported the first and largest cohort of nine immunocompetent patients with relapsed/refractory PCNSL treated with CD19 CAR-T cells (tisa-cel and axi-cel), after at least two previous lines of therapy [68] (Table 1). Despite the recent identification of CD19-expressing mural cells surrounding the brain endothelium as potential off-tumor targets [69], the authors did not observe unexpected neurotoxicity. Responses were centrally reviewed, according to the IPCG criteria [70]. With a median follow-up of 6.5 months, the best response was PR in one patient and CR in five patients, which demonstrated an activity of CAR-T cells in this specific setting. Median PFS was 4 months in the whole group and 7 months in responder patients. Six-month OS was 89%. These encouraging results should be confirmed by prospective clinical trials. 

Bridging therapy between leukapheresis and CAR-T infusion may contribute to optimize the results of CAR-T. It is presumable that bridging therapy will be necessary for most of relapsed/refractory PCNSL patients because of the rapid evolution of the disease and the necessary time to manufacture CAR-T. Radiation therapy should be carefully assessed in this setting, as it has been reported feasible [71], and could be associated with a better outcome for relapsed/refractory large B-cell lymphoma treated with axi-cel [72].

### 3.5. Other Targeted Therapies

Some targeted therapies, such as Bruton’s tyrosine kinase (BTK) inhibitors or immunomodulatory drugs (IMIDs), may have an “on-target, off-tumor” effect on the PCNSL tumor microenvironment (Table 2).

Ibrutinib, the first-in-class BTK inhibitor, showed a favorable brain distribution through the BBB in preclinical mice models [73,74] and substantial activity in PCNSL was reported in retrospective [75] and early phase clinical trials [76,77]. A French group conducted a proof-of-concept phase II study with ibrutinib, a single agent, at 560 mg per day, until progression or unacceptable toxicity, in R/R PCNSL and PVRL [78]. Among 44 evaluable patients, the disease control rate after 2 months of continuous treatment (primary endpoint) was 70%, including 23% CR + uCR, 36% PR and 11% stable disease (SD). Responses were observed in all CNS compartments. The overall safety profile was good, albeit two patients who received concomitant corticosteroids, presented with pulmonary aspergillosis. After a median follow-up of 25.7 months, the median PFS was 4.8 months and the median OS was 19.2 months. Notwithstanding, the duration of response was higher than 12 months in 15 patients. Interestingly, no correlation was found between responses and mutations in the B-cell receptor (BCR) pathway, which were available for 18 patients. This suggests that, beyond BTK inhibition, ibrutinib could modulate the brain microenvironment and enhance local antitumor immune responses. Second generation BTK inhibitors were developed, namely, acalabrutinib, zanubrutinib, orelabrutinib and tirabrutinib and assessed in B-cell malignancies. To date, only tirabrutinib has been prospectively evaluated as a single agent in a phase I/II study, showing a favorable toxicity and efficacy profile in R/R PCNSL [79].

IMIDs have also been tested in relapsed or refractory PCNSL patients. A phase I study reported that lenalidomide penetrates the ventricular CSF and was associated with an overall response rate (ORR) of 64% as monotherapy in 14 patients with relapsed or refractory CNS lymphoma, including 6 PCNSL [80]. In the phase II REVRI study, R/R PCNSL (*n* = 34) and PVRL (*n* = 11) patients received an induction comprising eight cycles of the R^2^ regimen (rituximab + lenalidomide), followed by a 1-year maintenance of lenalidomide alone in responding patients [81]. At the end-of-induction, the ORR was 36%, including 29% CR/uCR. The R^2^ combination was active in all CNS compartments. Eighteen and five patients started and completed the maintenance phase, respectively. Four patients remained in CR at the end-of-treatment. The limited benefit observed here differed from a retrospective analysis of lenalidomide maintenance in 10 relapsed PCNSL patients [80]. With a median follow-up of 19.2 months, the median PFS and OS were 7.8 and 17.7 months, respectively. Manageable toxicities were reported, mostly hematological. An interesting analysis of the circulating immune cell populations showed that the blood CD4/CD8 ratio at baseline had a prognostic value in the REVRI study. Indeed, the median PFS was 9.5 months when the CD4/CD8 ratio was ≥ 1.6 versus 2.8 months. This finding highlighted the role played by the microenvironment regarding the response to IMIDs, which needs to be validated in an independent cohort. Another phase I study determined the maximal tolerated dose of pomalidomide as 5 mg in association with dexamethasone within a cohort of 25 R/R PCNSL and PVRL [82]. This combination resulted in an ORR of 48% with 32% CR/uCR and a median PFS of 5.3 months (Table 1).

**Table 1 cancers-13-05061-t001:** Summary of the main clinical data on PCNSL immunotherapies. ORR, overall response rate; CR, complete response; PFS, progression-free survival; alloHSCT, allogeneic hematopoietic stem cell transplantation; NR, not reached; vs., versus; NA, not available; EFS, event-free survival; HD-MTX, high-dose methotrexate; DoR, duration of response.

Treatment	N Patients	Median Follow-Up	ORR	CR	Median PFS	Comments
**AlloHSCT [37]**	6	45 months	4/6	4/6	NR	Retrospective study. Four patients alive in CR at 4 years
**Rituximab** [47] R-MBVP vs. MBVP (1st line)	199	32.9 months	81% vs. 75%	68% vs. 66%	NA	1-year EFS 52% vs. 49%
**Rituximab** [9] HD-MTX plus cytarabine vs. HD-MTX, cytarabine plus rituximab vs. HD-MTX, cytarabine, rituximab plus thiotepa (1st line)	227	30 months	53% vs. 74% vs. 87%	23% vs. 30% vs. 49%	NA	Long-term analysis: 7-year OS 37% for HD-MTX plus cytarabine plus rituximab vs. 26% for HD-MTX plus cytarabine independently of the consolidation arm [49]
**Nivolumab** [58]	5	17 months	5/5	4/5	NA	Results not confirmed in a prospective study (NCT02857426)
**Nivolumab** [59]	9	18 months	7/9	3/9	12 months	Results not confirmed in a prospective study (NCT02857426)
**Pembrolizumab [60]**	50	6.7 months	26%	16%	2.6 months	Median DoR 10 months
**CD19 CAR-T cells** [68]	9	6.5 months	6/9	5/9	4 months	Median DoR NR
**Ibrutinib [78]**	44	25.7 months	59%	23%	4.8 months	DoR > 12 months in 15 patients
**Tirabrutinib [79]**	44	9.1 months	64%	34%	2.9 months	
**Lenalidomide** [80]	14	NA	9/14	3/14	6 months	
**Lenalidomide plus rituximab** [81]	45	19.2 months	36%	29%	7.8 months	
**Pomalidomide plus dexamethasone** [82]	25	16.5 months	48%	32%	5.3 months	

**Table 2 cancers-13-05061-t002:** Potential “on-target, off-tumor” effects of ibrutinib and lenalidomide/pomalidomide.

Drug	Cellular Target		Potential Effect
**Ibrutinib**	Adaptive immunity	Lymphocytes	↑ Th1 immunity [83] ↑ Persistence [84] ↓ CD8+ T-cell exhaustion [85]
	Innate immunity	Myeloid-derived suppressor cells	↓Migration, depletion [86]
**IMIDs** **(lenalidomide/pomalidomide)**	Adaptive immunity	Lymphocytes	↑ Th1 immunity [87,88] ↑ Effector functions [89]
	Innate immunity	TAMs NK cells	↑ M1/M2 phenotype [90] ↑ Effector functions [91]

## 4. Future Perspectives

### 4.1. Combination Therapies

The significant results reported above sketch an optimistic landscape with new therapeutic combinations to be tested in PCNSL patients. Indeed, despite high antitumor activity in the first months of treatment, the duration of response remains short. This encourages the combination of treatments targeting different immune pathways. Such combinations may involve BTK inhibitors, ICI, IMIDs and/or CAR-T cells, and are described in Figure 2. A synergistic effect on the anti-lymphoma immune response can be expected from these combinations. For example, IMIDs-induced M1 polarization might be associated with an increased production of IFNγ and enhanced PD-L1 expression; thus, impairing T-cell antitumor functions. As such, a combination therapy of lenalidomide with an anti PD-L1 antibody could be beneficial. Ongoing clinical trials of combined therapies targeting immune pathways are reported in Table 3.

Conventional immunochemotherapy might also be optimized by targeting immune pathways. The results presented above call for a further assessment of ibrutinib and lenalidomide or pomalidomide in combination with immunochemotherapy. The ongoing phase II LOC-R01 trial (NCT04446962) randomizes ibrutinib versus lenalidomide, plus rituximab–methotrexate–procarbazine–vincristine (R-MPV) for the induction treatment in first-line PCNSL patients, younger than 60 years-old and eligible for autologous HSCT.

Finally, besides their direct antitumor effect and the immunomodulation of the PCNSL microenvironment, ibrutinib and IMIDs should be tested in combination with CAR-T. Both targeted therapies have been reported to improve the efficacy and safety of CD19 CAR-T in preclinical [92,93,94] and clinical [95] studies. Clinical trials, associated with high-quality biomarker studies, are warranted to evaluate the synergy between CAR-T and such immunomodulatory drugs in order to improve the prognosis of high-risk PCNSL patients.

### 4.2. Optimizing the Timing of Immunotherapy 

To date, the development of most new therapies is focused on chemo-refractory/relapsed PCNSL patients. However, relapses are associated with a poorer prognosis [8] and often worse patients’ fitness. Immunotherapies may be more effective if given earlier in the therapeutic strategy. Das et al. reported an extensive analysis on T-cell phenotypes of cells collected from the peripheral blood of children with solid tumors and lymphomas [96]. Interestingly, the percentage of naïve and central memory T-cells decreased, whereas terminal effectors increased along with cumulative chemotherapy cycles. Moreover, the in vitro expansion capacity of these T-cells significantly declined over time and increased the number of chemotherapy cycles. These data provide a rational basis to introduce immunotherapies as early as possible in the course of the disease. CD19 CAR-T cells are currently being evaluated in high-risk systemic large B-cell lymphoma with suboptimal response to first line therapy (ZUMA-12 study). An intermediate analysis was recently reported and showed a higher median number of naïve T-cells in the final CAR-T product and a higher peak expansion of CAR-T, as compared to patients who received at least two prior lines of immunochemotherapy [97].

ICI were tested in first line treatment for multiple solid tumor indications and showed impressive results [98,99,100]. Overall, immunotherapies which rely on the activation of T-cell antitumor immunity, should be evaluated in the first lines of PCNSL treatment, when endogenous-T cells are prone to better anti-lymphoma effects. Thus, one may expect that the future induction therapy for newly diagnosed PCNSL would include high-dose methotrexate, in combination with ICI and an immunomodulatory drug. CAR-T cells should be evaluated at first relapse and as a consolidation therapy in patients ineligible for ASCT. 

### 4.3. Improving Trafficking

Many hypotheses may explain the modest results observed with rituximab for PCNSL treatment, including the question of poor CNS diffusion and the existence of complement-dependent cytotoxicity, antibody-dependent cellular cytotoxicity and antibody-dependent cellular phagocytosis in the brain. Two phase I trials evaluated the intraventricular injection of rituximab, either as monotherapy [101], or in combination with methotrexate [102]. Intrathecal rituximab was safe at 10 and 25 mg and modest clinical responses were observed in refractory CNS lymphoma. Interestingly, an ancillary study showed the activation of the complement cascade within the brain microenvironment after an intraventricular administration of rituximab [103].

The trafficking of CAR-T to the CNS may also be challenging. Preclinical studies showed enhanced CAR-T activity following local delivery. Mulazzani et al. developed in vivo microscopy in an orthotopic murine model of PCNSL to show that intracerebral, rather than intravenous, injection of CD19 CAR-T resulted in a deeper infiltration and an increased control of the tumor growth [104]. Interestingly, following intracerebral injection, CAR-T persisted in the brain and the blood for up to 159 days, even after a complete regression of the CNS lymphoma. The superiority of intraventricularly injected CD19 CAR-T was recently corroborated in another preclinical study [105]. Importantly, the models developed in both studies were immunodeficient mice, which may hinder the translation to human disease. Notably, due to the lack of circulating human B-cells, CAR-T cells do not rapidly encounter their target once infused intravenously and this may impair their expansion. Ideally, these promising findings should be validated in preclinical studies using immunocompetent models [106]. An intraventricular delivery of CD19 CAR-T via an Ommaya reservoir, following the failure of intravenous infusion, is currently being evaluated in the CAROUSEL trial (NCT04443829). This clinical study also addresses the question of dose reduction when CAR-T are injected directly into the tumor region.

## 5. Conclusions

A better understanding of the origin and characteristics of the CNS immune cells, along with the growing amount of preclinical and clinical data on immunotherapies in B-cell malignancies, may lead to the development of therapeutic avenues to improve the prognosis of PCNSL.

## Figures and Tables

**Figure 1 cancers-13-05061-f001:**
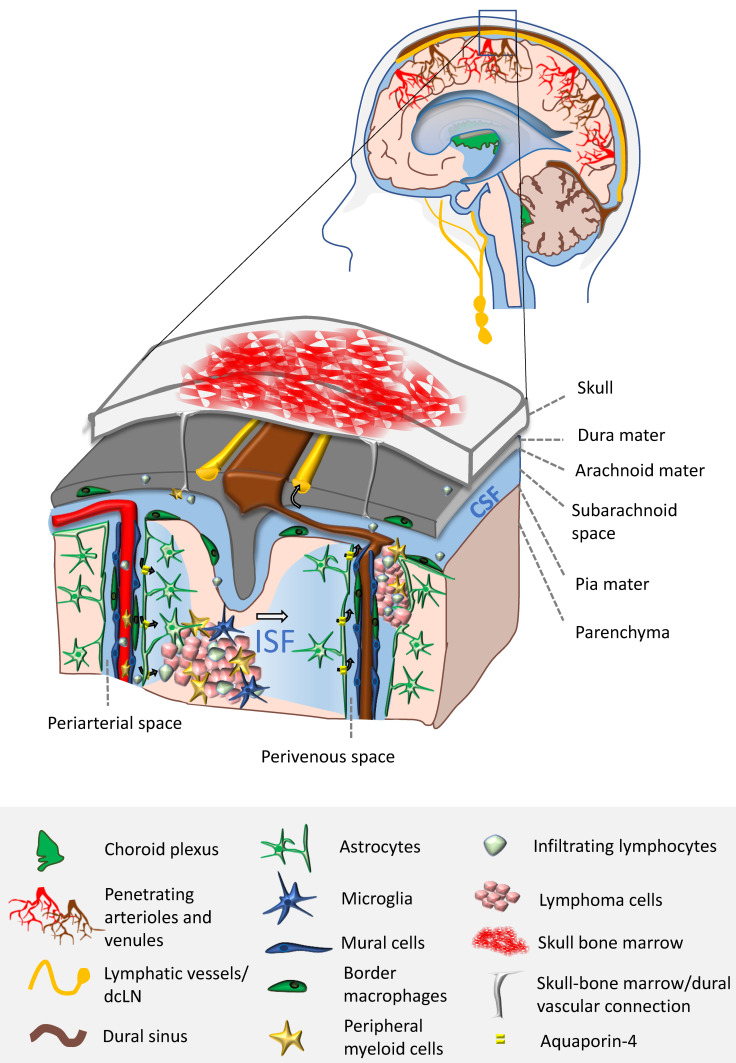
The TME in PCNSL is driven by the unique immune landscape and properties of the CNS. PCNSL are tumors that develop in alternative locations within the CNS. This has an impact on the TME cellular composition. When developing in cerebrospinal fluid (CSF) compartments (perivascular and meningeal spaces), lymphoma cells interact directly with border macrophages, lymphocytes, the glia limitans (formed by astrocytic endfeet), endothelial cells and mural cells (pericytes and smooth muscle cells). Inside the CNS parenchyma, tumor cells are in close contact with microglia, astrocytes and infiltrating immune cells: lymphocytes and peripheral myeloid cells. There are three potential sources of immune cells within PCNSL TME: derived from resident populations, from the blood and also from skull bone marrow reservoirs. Very recently, direct vascular connections between the skull bone marrow and the dura mater were found in mice. Antigens and immune cells from the TME of PCNSL are drained from the CSF compartment into deep cervical lymph nodes (dcLN) through meningeal lymphatic vessels, to potentially elicit anti-tumor responses. It is the role of the glymphatic system to clear CNS solutes, carrying antigens to the CSF compartment. Arrows indicate the directionality of CSF/Interstitial fluid (ISF) bulk flow, which is facilitated by Aquaporin-4 expressed on astrocytes.

**Figure 2 cancers-13-05061-f002:**
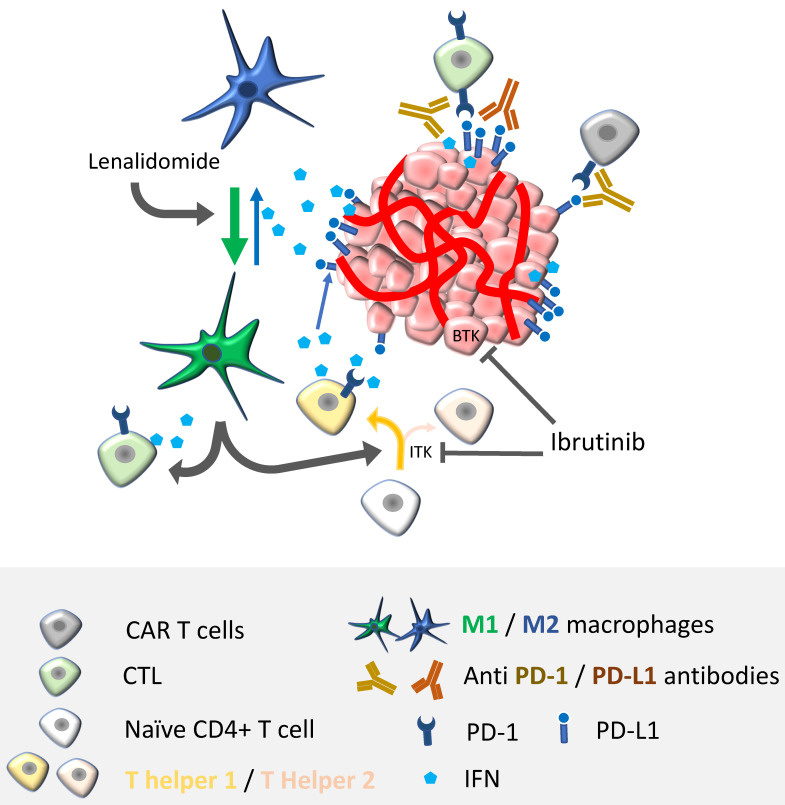
Perspectives for combination therapy in PCNSL. Besides just adding the effects from different treatments, one of the goals of combination immunotherapies is to find synergistic antitumor effects. Most of the direct antitumor activity is driven by antigen-specific or redirected T-cells (i.e., CAR-T cells). Therefore, unleashing antitumor response with immune checkpoint inhibitors should potentiate not only CAR-T cell responses, but also the effects of BTK/ITK inhibitors and immunomodulatory drugs, such as ibrutinib and lenalidomide/pomalidomide, respectively. Ibrutinib targets lymphoma cells by inhibiting BTK. Additionally, it regulates T-helper responses by limiting Th2 activation and inducing a shift in the Th2/Th1 ratio by targeting interleukin-2-inducible T-cell kinase (ITK). Lenalidomide/pomalidomide, by skewing M2-like macrophages towards M1 phenotypes, indirectly boosts cytotoxic T lymphocytes (CTL) activity and Th1 responses; therefore, enhancing IFN production, a cytokine that is well known for upregulating PD-L1 on tumor cells. Enhancing the adaptive antitumor immunity by using ibrutinib or lenalidomide/pomalidomide should, directly or indirectly, potentiate CAR-T cell functions.

**Table 3 cancers-13-05061-t003:** Ongoing clinical trials of combined therapies targeting immune pathways. PCNSL: primary central nervous system lymphoma; sCNSL: secondary central nervous system lymphoma; R/R: relapsed or refractory; MTD: maximal tolerated dose; ORR: overall response rate; CR: complete response; PFS: progression-free survival.

Clinicaltrials.gov Identification	Study Design	Treatment	Objective	Status
NCT04609046	Phase I PCNSL first-line	Induction: methotrexate, rituximab, lenalidomide and nivolumab Maintenance: lenalidomide and nivolumab		Ongoing (estimated enrolment: 27 patients)
NCT03703167	Phase Ib R/R PCNSL and R/R sCNSL	Combination of ibrutinib with rituximab and lenalidomide with dose expansion of ibrutinib and lenalidomide	MTD of ibrutinib PFS	Ongoing (estimated enrolment: 40 patients)
NCT04938297	Phase II PCNSL and sCNSL	Rituximab, zanubrutinib in combination with lenalidomide, followed by zanubrutinib or lenalidomide maintenance	ORR	Ongoing (estimated enrolment: 100 patients)
NCT04899427	Phase II R/R PCNSL	Orelabrutinib combined with PD-1 inhibitor	ORR	Ongoing (estimated enrolment: 32 patients)
NCT04831658	Phase II PCNSL first-line	BTK inhibitor, PD-1 inhibitor and formustine	CR rate	Ongoing (estimated enrolment: 40 patients)
NCT04737889	Phase II PCNSL	Rituximab, lenalidomide combined with methotrexate and temozolomide	2-year PFS	Ongoing (estimated enrolment: 30 patients)
NCT04688151	Phase I PCNSL	Rituximab, acalabrutinib and durvalumab (RAD)	MTD Acalabrutinib	Ongoing
NCT04462328	Phase I PCNSL and sCNSL R/R and first-line	Dose expansion of acalabrutinib and durvalumab	MTD Acalabrutinib	Ongoing (estimated enrolment: 21 patients)
NCT04421560	Phase Ib/II R/R PCNSL	Pembrolizumab, ibrutinib and rituximab	6-month PFS	Ongoing (estimated enrolment: 37 patients)
NCT03770416	Phase I R/R PCNSL and sCNSL	Nivolumab and ibrutinib	ORR	Ongoing (estimated enrolment: 40 patients)
NCT04446962	Phase Ib/II PCNSL first-line	Lenalidomide or ibrutinib in association with rituximab–methotrexate–procarbazine–vincristine (R-MPV)	MTD lenalidomide/ibrutinib CR rate at the end of induction	Ongoing (estimated enrolment: 92 patients in phase II)

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
