# Peer review of "Emerging Landscape of Immunotherapy for Primary Central Nervous System Lymphoma"

_cancers, 2021, doi:10.3390/cancers13205061_

Round 1
Reviewer 1 Report
This review provided perspectives for efficacy improvement of select immunotherapies such as alloHSCT, PDL1, and CAR-T. Moreover, it asserted that the therapeutic status of PCNSL differs from that of non-CNS DLBCL owing to physiological barrier and molecular background.
The authors also discussed the tumor microenvironment based on recent findings. Although multidrug chemotherapy is the standard treatment for PCNSL, new treatments are needed to improve prognosis among affected patients, which is also a very important consideration from this perspective.
There are a few points that require clarifications.
- Please mention ASCT in the alloHSCT section to better understand the role of hematopoietic stem cell transplantation for PCNSL. Besides radiotherapy and standard chemotherapeutic agents, intensive chemotherapy with hematopoietic reconstruction and ASCT results in less detrimental effects on neurocognitive functions. However, elderly patients are not eligible for hematopoietic stem cell transplants. In addition, please describe the advantages and disadvantages of hematopoietic stem cell transplantation.
- In the BTK inhibitor section, please add the second generation of BTKis, including zanubrutinib, acalabrutinib, and orelabrutinib, to illustrate the evolutionary development of BTKis.
- In section 4.2., please add a discussion the new perspectives for appropriate schedule for combination HD-MTX, immune checkpoint inhibitors, BTKis, iMiDs, and CAR-T in the induction or consolidation phase.
Author Response
Reviewer 1
Open Review
(x) I would not like to sign my review report
( ) I would like to sign my review report
English language and style
( ) Extensive editing of English language and style required
( ) Moderate English changes required
(x) English language and style are fine/minor spell check required
( ) I don't feel qualified to judge about the English language and style
|
Is the work a significant contribution to the field? |
|
|
Is the work well organized and comprehensively described? |
|
|
Is the work scientifically sound and not misleading? |
|
|
Are there appropriate and adequate references to related and previous work? |
|
|
Is the English used correct and readable? |
Comments and Suggestions for Authors
This review provided perspectives for efficacy improvement of select immunotherapies such as alloHSCT, PDL1, and CAR-T. Moreover, it asserted that the therapeutic status of PCNSL differs from that of non-CNS DLBCL owing to physiological barrier and molecular background.
The authors also discussed the tumor microenvironment based on recent findings. Although multidrug chemotherapy is the standard treatment for PCNSL, new treatments are needed to improve prognosis among affected patients, which is also a very important consideration from this perspective.
We thank the reviewer for this positive comment.
There are a few points that require clarifications.
- Please mention ASCT in the alloHSCT section to better understand the role of hematopoietic stem cell transplantation for PCNSL. Besides radiotherapy and standard chemotherapeutic agents, intensive chemotherapy with hematopoietic reconstruction and ASCT results in less detrimental effects on neurocognitive functions. However, elderly patients are not eligible for hematopoietic stem cell transplants. In addition, please describe the advantages and disadvantages of hematopoietic stem cell transplantation.
As recommended by the reviewer, we considered adding details about ASCT in the alloHSCT section. However, the resulting alloHSCT section turned unclear and confused. Instead, we chose to add information about ASCT within the introduction, and some informations about the objectives and the toxicity of alloHSCT in the alloHSCT section. The main objective of this section was to present the first immunotherapy tested in PCNSL. We did not enter into the details of the eligibility criteria for alloHSCT to better focus of the immunological concept of alloHSCT and on the few available data of alloHSCT in PCNSL.
- In the BTK inhibitor section, please add the second generation of BTKis, including zanubrutinib, acalabrutinib, and orelabrutinib, to illustrate the evolutionary development of BTKis.
We agree with the reviewer that these data were missing and we added it in the corresponding section, including the reference for the study by Narita et al, Neuro Oncol 2021 with Tirabrutinib single agent in R/R PCNSL.
- In section 4.2., please add a discussion the new perspectives for appropriate schedule for combination HD-MTX, immune checkpoint inhibitors, BTKis, iMiDs, and CAR-T in the induction or consolidation phase.
We thank the reviewer for this suggestion, which reinforce our message. Such discussion has been added to the corresponding section.
Reviewer 2 Report
Well written comprehensive review of immunotherapy for PCNSL addressing both preclinical and clinical data. Nice job describing results of multiple small studies that are useful a very useful summary of recent approaches in this rare type of lymphoma. This data is helpful for treating MDs despite the lack of large or randomized trials. The brain microenvironment section and Figure 1 are excellent.
Minor comments.
- Consider a sentence or two providing more detail re the IL-10 data so reader does not have to look it up. Not well known among clinicians.
- Is there any preliminary data of the Nivolumab trial for R/R CNS lymphoma. The small case series stimulated everyone's interest in this approach several years ago and a large trial was initiated but never published?? Were there toxicities that stopped the trial? Concerning that this data is not available. Pembro data disappointing. Should at least mention the nivo study re number of pts and when closed and mentioned never published.
- "expectable" not in common use - consider expected.
- Medawar's
- Page 3 line 106 should be "In light of" - not with
- Page 5 line 164 consider replacing input with inclusion.
- Page 6, line 205 consider replacing "slight level of" with modest
- NCT0468815 is now ongoing.
Author Response
Reviewer 2
Open Review
(x) I would not like to sign my review report
( ) I would like to sign my review report
English language and style
( ) Extensive editing of English language and style required
( ) Moderate English changes required
(x) English language and style are fine/minor spell check required
( ) I don't feel qualified to judge about the English language and style
|
Is the work a significant contribution to the field? |
|
|
Is the work well organized and comprehensively described? |
|
|
Is the work scientifically sound and not misleading? |
|
|
Are there appropriate and adequate references to related and previous work? |
|
|
Is the English used correct and readable? |
Comments and Suggestions for Authors
Well written comprehensive review of immunotherapy for PCNSL addressing both preclinical and clinical data. Nice job describing results of multiple small studies that are useful a very useful summary of recent approaches in this rare type of lymphoma. This data is helpful for treating MDs despite the lack of large or randomized trials. The brain microenvironment section and Figure 1 are excellent.
We are grateful to the reviewer for this kind comment.
Minor comments.
- Consider a sentence or two providing more detail re the IL-10 data so reader does not have to look it up. Not well known among clinicians.
We added more informations about IL-10 in section 2 (brain microenvironment).
- Is there any preliminary data of the Nivolumab trial for R/R CNS lymphoma. The small case series stimulated everyone's interest in this approach several years ago and a large trial was initiated but never published?? Were there toxicities that stopped the trial? Concerning that this data is not available. Pembro data disappointing. Should at least mention the nivo study re number of pts and when closed and mentioned never published.
We agree with the reviewer that the first published small case series with nivolumab generated great enthusiasm. Unfortunately, the nivolumab trial for R/R CNS lymphoma was disappointing and did not reproduce the results of the case series (in which 2/5 patients had received WBRT just prior to the introduction of nivolumab). Some data are available on clinicaltrials.gov. We added a discussion about the nivolumab study in the corresponding section of the manuscript.
- "expectable" not in common use - consider expected. Done
- Medawar's. Revised
- Page 3 line 106 should be "In light of" - not with. Revised
- Page 5 line 164 consider replacing input with inclusion. We replaced input with addition
- Page 6, line 205 consider replacing "slight level of" with modest. Revised
8. NCT0468815 is now ongoing. Revised in table 2. Thank you for the updated information.
Reviewer 3 Report
In this review, Alcantara et al. provides a very comprehensive overview of our current understanding of the brain microenvironment and mechanisms of potential immune escape in PCNSL particularly the expression of immune checkpoint receptors and PD-L1 on tumor-infiltrating lymphocytes and tumor-associated macrophages. The authors then provide an excellent review of the role of targeted agents, CAR T-cells and immune checkpoint inhibitors in patients with PCNSL, as well as future perspectives on how combination therapies of these agents may provide synergistic anti-tumor effects.
Overall, the topic is of interest in the current era of novel immunotherapies, is concise, yet comprehensive. I have no major concerns other than some suggested edits:
- Under the section on CAR-T cells, it was mentioned in line 255 that the hypothesis that the immunosuppressive TME may have contributed to lymphoma relapse in those that received CAR T-cells and that this should be followed-up with more preclinical and clinical studies. It would be helpful for the authors to specify what particular studies they think would be helpful or to speculate on what aspects of the immunosuppressive TME could be contributing to lack of significant responses to CAR T-cells in the PCNSL population.
- Figure 2 provides a great overview of the various mechanisms of action on various treatments for PCNSL and how they may work synergistically. However, the figure itself is a little hard to visualize as the labels for various parts of the image are scattered throughout. Would be helpful to have a quick image key (i.e. for PD-L1, PD1, CART-cells, etc).
- Would recommend review for grammatical errors and word choice. For example, on lines 72, 126, and 356, did the authors mean “efficacy” instead of “efficiency”? Other examples: line 144 (“traffic the CNS)”, line 153 (“despite preliminary”), line 157 (“positioning regarding other cell-therapies”), line 164 (“input of rituximab”), lines 182-187, line 266 (“assumable”), line 265, line 270, line 316-317 ( “association of therapies”).
Author Response
Reviewer 3
Open Review
(x) I would not like to sign my review report
( ) I would like to sign my review report
English language and style
( ) Extensive editing of English language and style required
(x) Moderate English changes required
( ) English language and style are fine/minor spell check required
( ) I don't feel qualified to judge about the English language and style
|
Is the work a significant contribution to the field? |
|
|
Is the work well organized and comprehensively described? |
|
|
Is the work scientifically sound and not misleading? |
|
|
Are there appropriate and adequate references to related and previous work? |
|
|
Is the English used correct and readable? |
Comments and Suggestions for Authors
In this review, Alcantara et al. provides a very comprehensive overview of our current understanding of the brain microenvironment and mechanisms of potential immune escape in PCNSL particularly the expression of immune checkpoint receptors and PD-L1 on tumor-infiltrating lymphocytes and tumor-associated macrophages. The authors then provide an excellent review of the role of targeted agents, CAR T-cells and immune checkpoint inhibitors in patients with PCNSL, as well as future perspectives on how combination therapies of these agents may provide synergistic anti-tumor effects.
Overall, the topic is of interest in the current era of novel immunotherapies, is concise, yet comprehensive. I have no major concerns other than some suggested edits:
We warmly thank the reviewer for its positive feedback on our manuscript.
- Under the section on CAR-T cells, it was mentioned in line 255 that the hypothesis that the immunosuppressive TME may have contributed to lymphoma relapse in those that received CAR T-cells and that this should be followed-up with more preclinical and clinical studies. It would be helpful for the authors to specify what particular studies they think would be helpful or to speculate on what aspects of the immunosuppressive TME could be contributing to lack of significant responses to CAR T-cells in the PCNSL population.
We thank the reviewer for this suggestion and we detailed some ideas that we feel relevant in this setting.
- Figure 2 provides a great overview of the various mechanisms of action on various treatments for PCNSL and how they may work synergistically. However, the figure itself is a little hard to visualize as the labels for various parts of the image are scattered throughout. Would be helpful to have a quick image key (i.e. for PD-L1, PD1, CART-cells, etc).
As suggested, the figure has been modified to facilitate visualization.
- Would recommend review for grammatical errors and word choice. For example, on lines 72, 126, and 356, did the authors mean “efficacy” instead of “efficiency”? Other examples: line 144 (“traffic the CNS)”, line 153 (“despite preliminary”), line 157 (“positioning regarding other cell-therapies”), line 164 (“input of rituximab”), lines 182-187, line 266 (“assumable”), line 265, line 270, line 316-317 ( “association of therapies”).
The manuscript has been revised in line of these comments.
Reviewer 4 Report
The work by Alcantara et al is about the usage of immunotherapy for the treatment of PCNSL patients. The topic is interesting, nevertheless I have some comments:
1) Introduction: why a considerable proportion of patients are refractory to high-dose methotrexate? What is the mechanism for that? Can authors specify the meaning for "a considerable proportion?
2)Introduction: info about the epidemiology have to be added
3) A table summarizing the available clinical data would be of help for readers
4) ICIS: authors report the possibility of future studies in combination therapy. Can authors be more specific?
5) CART-T cells: authors should report more details about the reported studies such as inclusion criteria, how patients have been checked during the follow up and how the treatment response has been evaluated. Also what is the limitation of this therapy for PCNSL patients? How to overcome the limitations?
6) Lenalidomide, Temozolomide have not been reported
7) What is the role of TILs in PCNSL patients?
Author Response
Reviewer 4
Open Review
(x) I would not like to sign my review report
( ) I would like to sign my review report
English language and style
( ) Extensive editing of English language and style required
( ) Moderate English changes required
(x) English language and style are fine/minor spell check required
( ) I don't feel qualified to judge about the English language and style
|
Is the work a significant contribution to the field? |
|
|
Is the work well organized and comprehensively described? |
|
|
Is the work scientifically sound and not misleading? |
|
|
Are there appropriate and adequate references to related and previous work? |
|
|
Is the English used correct and readable? |
Comments and Suggestions for Authors
The work by Alcantara et al is about the usage of immunotherapy for the treatment of PCNSL patients. The topic is interesting, nevertheless I have some comments:
1) Introduction: why a considerable proportion of patients are refractory to high-dose methotrexate? What is the mechanism for that? Can authors specify the meaning for "a considerable proportion?
Sixteen to 26% of patients are primary refractory to high-dose methotrexate or subsequently relapse, and we specified it in the introduction.
Unfortunately, as far of our knowledge, we don’t know the underlying mechanisms of primary refractory disease. Several mechanisms could be suggested such as i) the heterogeneity of malignant lymphoma clones, ii) pharmacogenomics specificity of methotrexate transport into the brain, and iii) the heterogeneity of the blood-brain barrier permeability around the lymphoma cell infiltration etc. However, none of these hypotheses have been proven yet.
2)Introduction: info about the epidemiology have to be added
Such informations were added to the introduction.
3) A table summarizing the available clinical data would be of help for readers
We thank the reviewer for this suggestion. Table 1 has been added to the manuscript and summarizes the main clinical data on PCNSL immunotherapies.
4) ICIS: authors report the possibility of future studies in combination therapy. Can authors be more specific?
We thank the reviewer for this suggestion. We have reinforced the discussion about combination therapy in section 4.2.
5) CART-T cells: authors should report more details about the reported studies such as inclusion criteria, how patients have been checked during the follow up and how the treatment response has been evaluated. Also what is the limitation of this therapy for PCNSL patients? How to overcome the limitations?
We thank the reviewer for this comment. We added more details about inclusion criteria and response evaluation on the largest cohort of PCNSL patients treated with CAR-T cells, reported by the LOC network.
To our knowledge, the main limitations of CAR-T therapy for PCNSL patients relied on efficacy and safety concerns. First, CAR-T trafficking to the brain tumor following intravenous infusion had to be assessed. Then, patients with PCNSL were excluded from the pivotal trials mainly because of safety concerns related to ICANS after CAR-T therapy. Nonetheless, these limitations do not longer seem relevant regarding the available data reported in our manuscript. We strongly believe that CAR-T therapy should be evaluated widely in clinical trials dedicated to PCNSL patients.
6) Lenalidomide, Temozolomide have not been reported
Lenalidomide is reported in the second part of section 3.5, dedicated to IMIDs. As an alkylating agent, temozolomide has not been reported in this manuscript, which focuses on immunotherapies for PCNSL.
7) What is the role of TILs in PCNSL patients?
As far of our knowledge, what is currently known about the role of TILs in PCNSL has been discussed in section 2 “brain microenvironment”.
Round 2
Reviewer 4 Report
I suggest the MS for publication